# Electroacupuncture and Manual Acupuncture Increase Joint Flexibility but Reduce Muscle Strength

**DOI:** 10.3390/healthcare8040414

**Published:** 2020-10-20

**Authors:** Daeho Kim, Sein Jang, Jihong Park

**Affiliations:** 1Department of Sports Science and Rehabilitation, Woosong University, Daejeon 34606, Korea; daeho.kim@wsu.ac.kr; 2Bareun Korean Medicine Clinic, Seoul 05616, Korea; jangsein@gmail.com; 3Department of Sports Medicine, Athletic Training Laboratory, Kyung Hee University, Yongin 17104, Korea

**Keywords:** range of motion, joint position sense, quadriceps, central activation ratio

## Abstract

The objective of this study was to investigate the immediate effects of electroacupuncture and manual acupuncture on hip flexion range of motion (ROM), knee joint (flexion replication at 15° and 45°) and quadriceps (strength and activation) function. Forty-five neurologically healthy adults participated in this randomized controlled laboratory study. Straight leg raise test, modified Thomas test, and hip abductors strength test were performed to determine acupoints. Afterwards, one of three 15-min treatments (control—no treatment, electroacupuncture, or manual acupuncture) was randomly applied using determined acupoints. Measurements (hip flexion ROM, and knee joint and quadriceps function) were recorded at baseline, and at 0, 20, and 40 min post treatment. Both electroacupuncture (4.0°, ES = 0.41) and manual acupuncture (5.4°, ES = 0.95) treatment immediately increased hip flexion ROM, and the increased values persisted for 40-min (*p* = 0.01). Knee flexion replication (at 15°: *p* = 0.17; 45°: *p* = 0.19) and quadriceps activation (*p* = 0.71) did not change at any of the time points. Post-treatment, both electroacupuncture and manual acupuncture decreased quadriceps strength at 0-min (electroacupuncture: 9.2%, *p* < 0.0001, ES = 0.60) and 40-min (electroacupuncture: 7.3%, *p* = 0.005, ES = 0.55; manual acupuncture: 8.7%, *p* = 0.01, ES = 0.54). A single session of either electroacupuncture or manual acupuncture treatment (selected acupoints based on physical examination) may immediately improve joint flexibility but reduce muscle strength.

## 1. Introduction

Along with its popularity for pain relief for over 2500 years [1], acupuncture has also been shown to improve joint [2,3,4] and muscle [5,6,7] function. For example, an application of electroacupuncture applied to five acupoints (GB34, SP9, SP10, SP34, and ST36) resulted in an increase of knee and hip joint mobility during walking [4] and six acupoints (GB20, BL10, BL43, TE15, SI13, and GV14) of manual acupuncture [3] improved cervical joint mobility (rotation and lateral flexion). It is believed that acupuncture needles induce a mechanical effect for muscular relaxation [2] and intervention for reducing muscle spasticity [8], which in turn, increases muscular extensibility and joint mobility. Regarding muscular function, two acupoints (ST36 and ST39) of electroacupuncture or manual acupuncture improved ankle dorsiflexion strength [6] and four acupoints (ST36, SP6, REN6, and ear point 55) of manual acupuncture increased rectus femoris activation [5]. Although it is not fully understood, the insertion and manipulation of the metal needles are thought to cause alterations in afferent sensory input to the central nervous system, resulting in an increase in α motoneuron excitability [9].

Whereas, metal acupuncture needles play crucial roles in producing mechanical and/or neurophysiological effects on functional improvements, an additional application of electrical stimulation (e.g., transcutaneous electrical nerve stimulation: TENS) may theoretically produce a more positive effect. Since the combined treatment form of acupuncture and electrotherapy was first introduced in 1820 [10], many studies [6,11,12,13,14,15,16] have compared the effectiveness between electroacupuncture and manual acupuncture. In regards to pain, electroacupuncture was effective in reducing pain in patients with elbow [14], lower back [15], and hand [16] conditions. In regards to function, there were no differences in knee functional mobility [11] and strength in knee extension [12] for knee osteoarthritis, and strength in ankle plantarflexion [13] and ankle dorsiflexion [6] in healthy population. While, electroacupuncture is considered a superior treatment to manual acupuncture for pain, it is still a debate of controversy when compared functional outcomes.

In oriental medicine clinics, the selection of acupoints for a specific purpose is often based on the patient’s overall condition. For example, a different number of acupuncture needles on different acupoints can be inserted for individuals with the same pathology. However, previous studies have standardized this treatment option, which does not reflect actual practice in the field. For example, in previous studies comparing the effectiveness between electroacupuncture and manual acupuncture, the standardized (same) acupoints were used (GB34 and ST38 [11]; ST36 and ST39 [6]). Since musculoskeletal conditions, such as the level of joint flexibility and muscle strength vary within any population, individual differences in such conditions should take account of acupoint selections.

Therefore, the purpose of this study was to determine the immediate effects of electroacupuncture and manual acupuncture treatment compared with non-acupuncture conditions on hip flexion range of motion (ROM), knee joint function (knee flexion replication at 15° and 45°), and quadriceps function (strength and activation) in healthy people. Regardless of treatment conditions, acupoints for each participant in this study were determined based on the results of three physical examinations (modified Thomas test, straight leg raise test, and gluteal medius manual muscle test). We hypothesized that either type of acupuncture would immediately increase hip flexion ROM, knee joint function, and quadriceps function more than the no-acupuncture condition. We further hypothesized that electroacupuncture would increase hip flexion ROM, and quadriceps strength and activation more than manual acupuncture since TENS is known to produce relaxation of the mechanical properties of muscle tissue [17] and a disinhibitory effect on α-motoneuron [18]. The results of this study may determine which type of acupuncture is superior in terms of hip joint mobility and quadriceps function. In addition, the implementation of individualized acupoint selections would be evidence supporting a common clinic use.

## 2. Material and Methods

### 2.1. Participants

A total of 50 volunteers were initially scheduled for data collection. Three of them did not participate in the study due to schedule conflicts and two participants were excluded after the baseline assessments (Figure 1). Finally, forty-five neurologically healthy adults (age: 23.4 ± 3.0 years; height: 176.1 ± 5.1 cm; mass: 76.0 ± 9.2 kg) with no history of lower extremity and spine conditions resulting in surgery and no lower extremity and lower back injury in the past 6 months were included and analyzed. Prior to data collection, we obtained informed consent as approved by the University’s Institutional Review Board, which also approved the study. Participants also completed the lower extremity functional scale (LEFS) [19] to confirm that their lower extremity function was normal.

### 2.2. Study Design

A single blind randomized controlled laboratory study with repeated measurements over time. Independent variables were treatment (control: no treatment, electroacupuncture, and manual acupuncture) and time (baseline, 0-, 20-, and 40-min post treatment). Measurements and treatments were undisclosed. The order of randomization was determined by the 3 × 4 Latin Square design degenerated by a statistical package (SAS Institute Inc., Cary, NC, USA) and then randomized again with the random allocation function in a spreadsheet program (Excel, Microsoft Corp., Redmond, WA, USA).

### 2.3. Procedures

All participants visited the laboratory once and were dismissed after data collection. The flow of participants is illustrated in Figure 1.

Upon their arrival at the laboratory, participants took 10-min of rest in the supine position after filling out paperwork (informed consent and LEFS). Next, the tester performed the baseline measurements (hip flexion ROM, knee flexion replication at 15° and 45°, and quadriceps strength and and activation). Once the baseline values were recorded, the tester left and the acupuncturist came into the laboratory. After physical examination was performed for acupoint selection, subjects were randomly assigned to one of the three treatment options (control: no treatment, electroacupuncture, and manual acupuncture) using the opaque envelope method (Table 1). After the treatment, the tester, who were blinded to treatment, came back to the laboratory and were assessed at 0-, 20-, and 40-min post treatment (the same measurements and order as the baseline measurements).

### 2.4. Measurements

At each time point, hip flexion ROM, knee flexion replication at 15° and 45°, and quadriceps strength and activation were performed in the respective order. Three successful trials were recorded for all measurements.

To measure hip flexion ROM, participants were asked to raise their dominant leg straight, with their knee as fully extended as possible, and hold the position for 5 s. The assessor measured the hip flexion ROM using a goniometer (movable arm: the lateral malleolus of the ankle; axis: the greater trochanter of the femur; stationary arm: parallel to the trunk) [20].

For knee flexion replication at 15° and 45°, participants were seated blindfolded on a dynamometer (Cybex 770, Cybex Inc., Medway, MA, USA; sampling rate: 100 Hz) with their knee resting at 90° (set position). After strapping the lower leg was strapped, the participant’s leg was passively moved to 15° (angle A) and 45° (angle B) of knee flexion to cognize angle A and B [21]. Participants had 3 s of holding time to help them remember these two angles. Participants were then asked to move their leg to angle A and B (in a random order).

After the knee joint replication tests, the dynamometer was locked for quadriceps strength (knee at 90° and hip at 85° flexion). Participants were then asked to produce maximal knee extension. Once the quadriceps strength reached a plateau, a supramaximal exogenous electrical stimulus (100 pulses per second, 450-µ pulse duration, 10 trains in 100-ms duration, and 125 V with peak output current 450 mA) was manually delivered and transmitted directly to the quadriceps through the two stimulating electrodes to evoke superimposed burst (SIB) force. Quadriceps activation was calculated as quadriceps force/SIB force. This activity was performed with 30-s rest intervals. A S88 Grass Stimulator with a SIU8T transformer stimulus isolation unit (Grass-Telefactor, West War-wick, RI, USA) was used to generate supramaximal exogenous electrical stimulus. Two electrodes (Dura-Stick II; Chattanooga, Hixson, TN, USA; 70 × 127 mm) were attached at the proximal lateral aspect and the distal medial aspect.

### 2.5. Physical Examination

A licensed acupuncturist performed a physical examination of strength and flexibility to determine a customized acupuncture treatment for each patient (Table 1). Strength and flexibility were tested to address how much of each participant’s movement patterns [22]. Two tests were performed to flexibility: Straight leg raise test [23] and modified Thomas test [24].

For the straight leg raise test (Figure 2A), participants were placed in a supine position on the table with arms-crossed over the chest. The acupuncturist grasped under the participant’s heel with one hand, while placing the other hand on the anterior knee to keep it in full extension during the examination. The acupuncturist raises the leg by flexing the hip until the participant felt discomfort or until the full ROM was obtained (Figure 2). We indicated a positive sign if the hip flexion ROM was less than the normal ROM (70°) [20]. Once participants received a positive sign on this test, option A was added to the acupuncture treatment (Table 1).

For the modified Thomas test (Figure 2B), participants were placed in a supine position with their knees bent at the end of the table. One leg was passively flexed towards the patient’s chest, allowing the knee to flex during the movement. The opposite leg (the leg being tested) rested flat on the table (Figure 2). We indicated a positive sign if the lower leg moved into extension or the involved leg rose off the table [20]. Once participants received a positive sign on this test, option B acupoints were added (Table 1).

For the gluteal medius manual muscle test (Figure 2C), participants lay on the opposite side of the muscle being tested with their knees slightly flexed on the table. Participants held a position of the dominant hip abduction at 30° with their knees fully extended while the pelvis and torso were actively stabilized. The acupuncturist gave resistance to the lateral femoral condyle for 5 s. We indicated a positive sign if the patient could not resist against the maximal pressure (5 out of 5 on manual muscle test) [20]. Once participants received a positive sign on this test, option C was added (Table 1).

### 2.6. Treatments

The acupuncturist (the same person as the physical examiner) performed a randomly assigned 15-min treatment (control: *n* = 13, electroacupuncture: *n* = 18, or manual acupuncture: *n* = 14). Participants were placed in a supine position on the treatment table (Figure 3). Disposable sterile stainless-steel acupuncture needles (0.3 × 40 mm; Dongbang Acupuncture, Seongnam, Korea) with a guide tube were used to minimize discomfort. The needles were inserted at a depth of 5–30 mm depending on the participant’s size and the conditions of the subcutaneous tissues. After the needle insertion, the needle was rotated in either clockwise or counterclockwise direction to make the participant feel a de qi sensation. The options of acupoints are presented in Table 1. Details of the acupuncture treatment were summarized according to the STRICTA [25] format (Table 2).

For the electroacupuncture condition, an electrical-stimulator (STN 110, StraTek Co., Anyang Korea) was used to provide alternating current with symmetrical bimodal pulse, continuous wave mode. The acupoints adjacent knee joint (a pair of SP9 and SP10; ST34 and ST36; GB31 and GB34: Figure 3) were selected to connect and close the circuit [26] to induce a facilitatory effect (e.g., above normal activation) on the quadriceps. The pulse frequency and duration were set at 170 Hz and 480 s (phase duration 55 μs) respectively. The intensity was set manually to the extent that there was no muscle contraction. For the manual acupuncture treatment, the same methods as for the electroacupuncture were applied except for the electrical stimulation. Participants remained in a supine position on the table for 15 min. They were repeatedly asked whether they felt muscle twitches, tingling, or burning sensations. The stimulus intensity was gradually increased until they reported a strong but well-tolerable sensation. Participants in the control condition were remained in a supine position on the table for 15 min.

### 2.7. Statistical Analysis

We performed a priori power analysis based on previous research to calculate the sample size. The expected mean difference in quadriceps activation (CAR) was 0.09 with a standard deviation of 0.11, which yielded an effect size of 0.82 [28]. We used an alpha of 0.05 and a beta of 0.2. These calculations estimated that 13 participants would be necessary in for each treatment.

Mean and 95% confidence interval values were calculated from each measurement in each time point. We performed Shapiro-Wilk test to determine parametric (mixed model analysis of variance) or non-parametric (Kruskal-Wallis) test on each dependent measurement. Tukey-Kramer tests were used for post-hoc pairwise comparisons. (We used a statistical package R for all tests (R 3.4.3; R Development Core Team, Vienna, Austria; *p* < 0.05). We also calculated effect sizes (ES = [$\bar{X}$_1_–$\bar{X}$_2_]/σ_pooled_) when significant between-time differences were observed.

## 3. Results

Electroacupuncture (4.0°, *p* = 0.0006, ES = 0.41) and manual acupuncture (5.4°, *p* < 0.0001, ES = 0.95) immediately increased hip flexion ROM (treatment × time interaction: F_6,126_ = 2.91, *p* = 0.01) at 0-min post treatment, and the increased values lasted until 40-min post treatment (electro: 4.5°, *p* < 0.0001, ES = 0.48; manual: 6.0°, *p* < 0.0001, ES = 1.1; Table 3). However, there was no distinctively superior treatment (treatment main effect: F_2,126_ = 1.85, *p* = 0.16). Regardless of the treatment (time main effect: F_3,126_ = 21.12, *p* < 0.0001), there was a 5.4% increase at 0-min (*p* < 0.0001, ES = 0.43), a 5.5% increase at 20-min (*p* < 0.0001, ES = 0.47), and a 5.4% increase at 40-min (*p* < 0.0001, ES = 0.47) post treatment compared to the baseline values in each treatment.

Replications at 15° (treatment × time interaction: F_6,126_ = 1.53, *p* = 0.17) and 45° (treatment × time interaction: F_6,126_ = 1.47, *p* = 0.19) did not change at any time point (Table 4).

Electroacupuncture (9.2%, *p* < 0.0001, ES = 0.60) decreased quadriceps strength at 0-min post treatment, and the decreased values stayed the same until 40-min post treatment (7.3%, *p* = 0.005, ES = 0.55; Table 4). Manual acupuncture (8.7%, *p* = 0.01, ES = 0.54) decreased quadriceps strength at 40-min post treatment. Regardless of the treatment (time main effect: F_3,126_ = 18.65, *p* < 0.0001), quadriceps strength showed a 5.9% decrease at 0-min (*p* < 0.0001, ES = 0.33), a 5.1% decrease at 20-min (*p* < 0.0001, ES = 0.29), and a 7.5% decrease at 40-min (*p* < 0.0001, ES = 0.42) post treatment in relation to the baseline.

Quadriceps activation did not change at any time point (χ^2^ = 8.21, df = 11, *p* = 0.70; Table 5).

## 4. Discussion

We examined the immediate effects of a single session of electroacupuncture or manual acupuncture, but were not able to determine a superior treatment given the influence of both acupuncture treatments on joint and muscle functions were similar. Acupuncturists and clinicians should be aware of an increase in hip flexion ROM and a reduction in quadriceps strength. Different acupoint options were applied to each participant: at least three acupoints (control: *n* = 3; electro: *n* = 7; and manual: *n* = 1), six acupoints (control: *n* = 7; electro: *n* = 8; and manual: *n* = 7), and up to nine acupoints (control: *n* = 3; electro: *n* = 3; and manual: *n* = 6; total of seven different acupoint options; Table 1). Physical examination (straight leg raise test, modified Thomas test, and gluteal medius manual muscle test) was performed to evaluate the participants’ physical characteristics.

A single session of either treatment using electroacupuncture or manual acupuncture increased hip flexion ROM and the increased values were maintained for 40-min; these results were consistent with the results in previous studies [2,29]. Manual acupuncture (five [2] and eight [29] standardized acupoints) improved hip (8.8°) [2] and knee flexion (10.3°) [29] ROM in patients with knee osteoarthritis. Increased ROM in our subjects could be explained by central and peripheral mechanisms. Centrally, threshold of discomfort perception during SLR could possibly be increased from a counterirritant effect due to needle insertions [30]. Previous studies have shown that there are at least two possible mechanisms. In the first mechanism, serotoninergic fibers, from the nucleus raphe magnus release encephalin, which hinders the transmission of nociceptive signals [31]. In the second mechanism, so called diffuse noxious inhibitory control, is the result of noxious stimuli applied to shortened muscles [31]. Regional effects may be attributed to an increase in tissue elasticity due to micro-stretching [32] and circulatory effect [33] in contracted fibers [2]. More specifically, needle insertion may stimulate release of vasoactive substances (e.g., substance P and bradykinin), resulting in vasodilation, and an increase in blood flow [34]. Furthermore, according to Langevin [35], there may be a relationship between the mechanisms of acupuncture treatment and the traditional meridians, which connect various parts of the body, reflecting interactions among blood vessel, nerve, and immune networks [36].

In this study, quadriceps strength decreased immediately after electroacupuncture (stimulating lateral femoral cutaneous nerve and femoral nerve). Among previous studies [5,6,7,37,38,39] concerning acupuncture treatments on muscle function, some results [7,37,39] are consistent with our results, while others [5,6,38] are not. Manual acupuncture, for instance, was found to decrease tibialis anterior muscle activity [37] and wrist flexor strength [7], but with no change in quadriceps activation. In contrast, acupuncture (both electro and manual) with two acupoints (ST36 and ST39; stimulating peroneal nerve) improved quadriceps strength of ankle dorsiflexion [6]. In one study, the local acupoints may have decreased muscle strength by influencing the reflex of the muscle loop [40], and the decrease in quadriceps strength is related to change in motor responses [7]. Our study used local acupoints, SP10, ST31, and ST34, on the quadriceps, which may have influenced the reflex loop of the quadriceps, resulting in decreased muscle strength. Another researcher stated that there may have been a cross-transfer effect induced by intramuscular needling [6]. Interestingly, there was a difference in quadriceps strength between electroacupuncture and manual acupuncture. It immediately decreased after electroacupuncture treatment, but only at 40-min post-treatment after manual acupuncture. This result indicates that electrical activity on quadriceps may have allowed relaxation, which may have extra effects through extended stimulation of nerves or muscles. Quadriceps activation showed no change at any time points in our study, consistent with previous study [39] in which quadriceps activation did not change after standardized acupuncture (five needles: SP9, SP10, ST36, GB36, and ah shi point). Our result indicate that needle penetration and manipulation may not be enough to influence sensory input at the supraspinal level.

Our data support the practice of selecting acupoints based on the results of physical examination. This examination should be given priority, and then the correct acupoints could obtain optimal therapeutic acupuncture effects. Previous studies have tried to demonstrate the effectiveness of acupuncture using the same acupuncture (LR1, LR3, LR8, REN2, and GB34 for hip ROM [2]; ST36 and ST39 for ankle strength [6]) without considering subjects’ physical condition. We concluded that physical examination should precede acupuncture treatment because individuals have different physical characteristics, in order to maximize the effectiveness of acupuncture treatment. We were interested in observing the acupuncture treatment effect on knee joint function with the replication test. Previous studies showed that electroacupuncture could improve the proprioception of the knee joint [41] and that there was an improvement in ankle joint replication after eight weeks of electroacupuncture [42]. Inconstantly, there were no alterations in joint function after either type of acupuncture treatment. This may be because we performed only a single 15-min session of acupuncture, which may not have been enough stimulation to improve knee joint function. We presume that acupuncture treatment needs to be continued for more than one session to enhance knee joint function.

Important limitations and assumptions should be noted. Our method to select acupoints was based on individual physical condition, which may have influenced the results. While, a larger sample size could resolve this issue, the number of subjects (justified by a priori sample size calculation using the measurement of quadriceps activation) in our study may not be adequate for strong generalizability. Future studies should attempt to obtain a larger sample to confirm the effect of our method. We could not control the different level of daily activities of the participants, since they have their own occupations, which may have influenced strength and joint function. We had no sham-acupuncture group to test the potential placebo effects. Previous data concerning the effects of sham acupuncture reported that both types of acupuncture showed a similar effect in pain reduction in patients with jaw pain [43] and fibromyalgia [44]. However, it has been reported that sham-acupuncture (e.g., superficial acupuncture) is not regarded as physiologically inert [45]. The duration of acupuncture treatment was only 15 min, which may not have been enough time to have positive effects on neurophysiological changes (e.g., afferent sensory input). We assume that conditions of acupuncture treatment (e.g., needle insertion depth and location), the level of warm-up prior to baseline measurements and the environmental condition (temperature and relative humidity) were not different across each participant. Last, our study did not involve a comparison with standardized acupoints, given that a comparison between customized and standardized acupoints was not the primary purpose. Therefore, the results of this study should not be taken to imply that using individualized acupoints (based on physical examination) is superior to using standardized acupoints.

## 5. Conclusions

Using customized acupoints based on physical examination, receiving a 15-min treatment of, either electroacupuncture or manual acupuncture appears to improve hip flexion ROM, but not for quadriceps strength and knee joint function in healthy young adults. In terms of functional change (hip flexion ROM, knee joint function, and quadriceps function), neither electroacupuncture nor manual acupuncture gave a clearly superior result. While, the use of customized acupoints in patients with the same pathological conditions has not been scientifically validated, our data support this common clinical practice.

## Figures and Tables

**Figure 1 healthcare-08-00414-f001:**
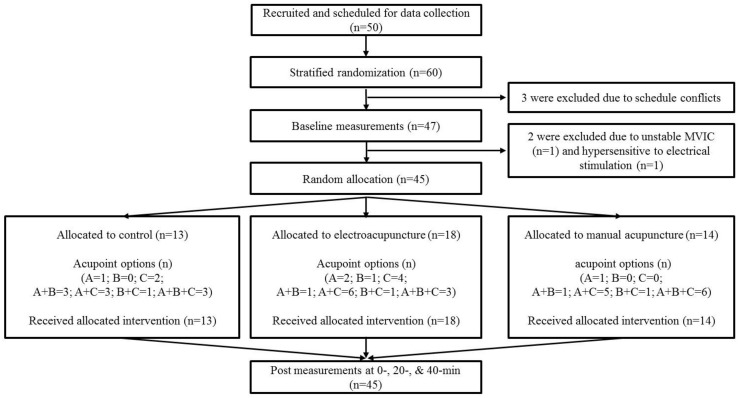
Flow diagram for the study. Ten additional participants were included when performing stratified randomization, based on considerations of drop-out rate. Acupoint option A: SP9, SP10, and SP11; B: ST31, ST34, and ST36; C: GB 30, GB 31, GB 34 (see Table 1 for more detailed information on acupoints).

**Figure 2 healthcare-08-00414-f002:**
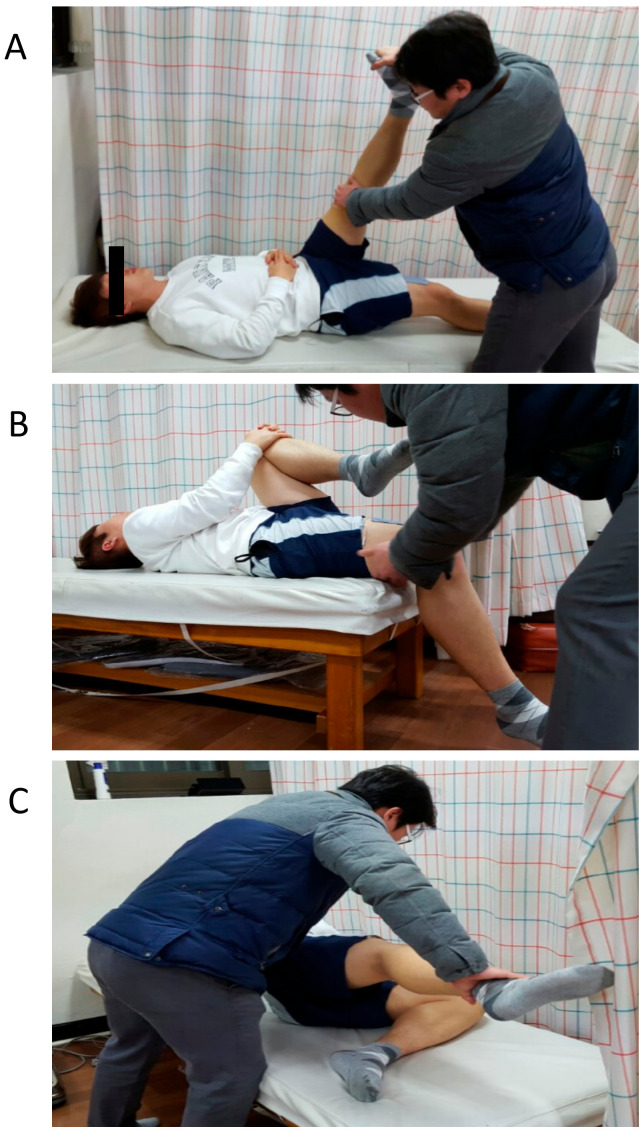
Physical examination. (**A**). straight leg raise test, (**B**). modified Thomas test, (**C**). gluteal medius manual muscle test.

**Figure 3 healthcare-08-00414-f003:**
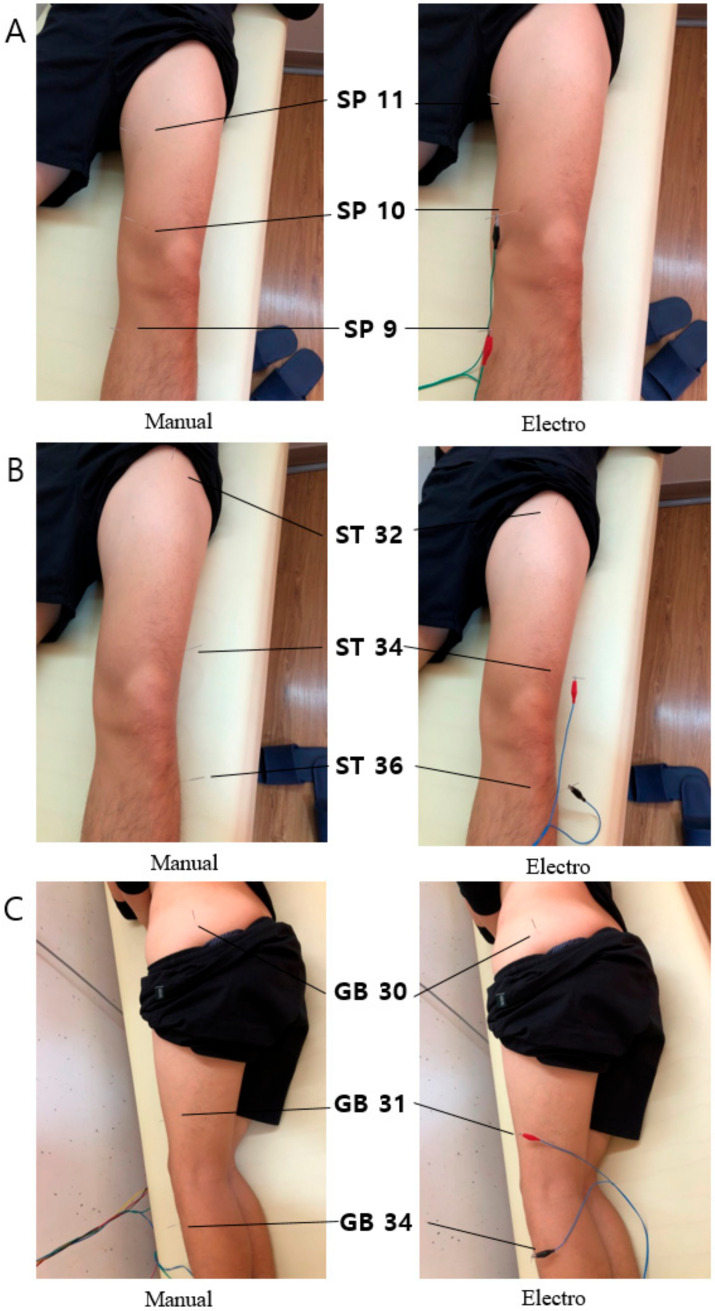
Acupoint options. The pictures in the left column are manual acupuncture and the right column are electroacupuncture. (**A**). SP9, SP10, and SP11, (**B**). ST32, ST34, and ST36, (**C**). GB30, GB31, GB34.

**Table 1 healthcare-08-00414-t001:** Acupoint options, number of participants, acupoints, targeted muscles, and anatomy trains (meridians).

Acupoint Options & Participants Number	Acupoints (# of Needles)	Targeted Muscles	Anatomy Trains (Meridian)
A (Control: 1; Electro: 2, Manual: 1)	SP9, SP10, and SP11 (3)	SP9 (popliteus and soleus), SP10 (vastus medius), and SP11 (adductor longus, magnus)	Deep front line (Spleen meridian)
B (Control: 0; Electro: 1, Manual: 0)	ST31, ST34, and ST36 (3)	ST31 (rectus femoris), ST34 (vastus lateralis), and ST36 (tibialis anterior and extensor digitorum longus)	Superficial front line (Stomach meridian)
C (Control: 2; Electro: 4, Manual: 0)	GB30, GB31, and GB34 (3)	GB30 (piriformis, gluteus medius, and gluteus minimus), GB31 (iliotibial band and vastus lateralis), and GB34 (peroneus longus and extensor digitorum longus)	Lateral line (Gallbladder meridian)
A + B (Control: 3; Electro: 1, Manual: 1)	SP9, SP10, SP11, ST31, ST34, and ST36 (6)	SP9 (popliteus and soleus), SP10 (vastus medius), SP11 (adductor longus, magnus), ST31 (rectus femoris), ST34 (vastus lateralis), and ST36 (tibialis anterior and extensor digitorum longus)	Deep front line (Spleen meridian) and Superficial front line (Stomach meridian)
A + C (Control: 3; Electro: 6, Manual: 5)	SP9, SP10, SP11, GB30, GB31, and GB34 (6)	SP9 (popliteus and soleus), SP10 (vastus medius), SP 11 (adductor longus, magnus), GB30 (piriformis, gluteus medius, and gluteus minimus), GB31 (iliotibial band and vastus lateralis), and GB34 (peroneus longus and extensor digitorum longus)	Deep front line (Spleen meridian) and Lateral line (Gallbladder meridian)
B + C (Control: 1; Electro: 1, Manual: 1)	ST31, ST34, ST36, GB30, GB31, and GB34 (6)	ST31 (rectus femoris), ST34 (vastus lateralis), ST36 (tibialis anterior and extensor digitorum longus), GB30 (piriformis, gluteus medius, and gluteus minimus), GB31 (iliotibial band and vastus lateralis), and GB34 (peroneus longus and extensor digitorum longus)	Superficial front line (Stomach meridian) and Lateral line (Gallbladder meridian)
A + B+C (Control: 3; Electro: 3, Manual: 6)	SP9, SP10, SP11, ST31, ST34, ST36, GB30, GB31, and GB34 (9)	SP9 (popliteus and soleus), SP10 (vastus medius), SP11 (adductor longus, magnus), ST31 (rectus femoris), ST34 (vastus lateralis), ST36 (tibialis anterior and extensor digitorum longus), GB30 (piriformis, gluteus medius, and gluteus minimus), GB31 (iliotibial band and vastus lateralis), and GB34 (peroneus longus and extensor digitorum longus)	Deep front line (Spleen meridian), Superficial front line (Stomach meridian), and Lateral line (Gallbladder meridian)
Total (Control: 13; Electroacupuncture: 18; Manual acupuncture: 14)			

The underlined acupoints are acupoints where electrodes are connected during electroacupuncture.

**Table 2 healthcare-08-00414-t002:** Treatment by the STRICTA Recommendation.

Item	Detail
1. Acupuncture rationale	(1a) Style of acupuncture: We selected the acupoints based on traditional Korean medicine (TKM) meridian theory [27] to improve the hip joint ROM and hip flexor tightness, and to treat gluteus medius weakness. (1b) Reasoning for treatment provided, based on physical examination: Acupoints were chosen for individual participants by physical examination (straight leg raise test, modified Thomas test, gluteal medius manual muscle test).
2. Details of needling	(2a) Number of needle inserts per subject: Three acupoints were inserted in case of positive in one physical examination. (2b) Names of points used (unilateral): Acupoints used were unilateral SP9, SP10 and SP11 points in case of positive in straight leg raise test (less than the normal ROM 70°), ST31, ST34, and ST36 points in case of positive in modified Thomas test, GB30, GB31, and GB34 points in case of positive in gluteal medius manual muscle test. (2c) Depth of insertion, based on a specified unit of measurement, or on a particular tissue level: In acupuncture treatment, the depth of needle insertion was about 5–30 mm. (2d) Manual acupuncture: It was performed for 15 min, the needle was rotated in either clockwise or counterclockwise direction to make the participant feel a de qi sensation. (2e) Electroacupuncture: It was performed for 15 min and electrically stimulation for 480 s (170 Hz, phase duration 55 μs within tolerable strength) using electro-stimulator (STN-110; StraTek Inc. Anyang, Korea). (2f) Needle type: The needles used for acupuncture were disposable stainless-steel needles (0.30 × 40 mm; Dong Bang Acupuncture., Seongnam, Korea).
3. Treatment regimen	(3a) Electroacupoints: Three acupoints were inserted in case of positive in one physical examination, two acupoints were stimulated just below and above the knee joint (SP9-SP10, ST34-ST36, and GB31-GB34). (3b) Electroacupuncture stimulation: It was delivered at an intensity that the participant could notice but felt comfortable with.
4. Practitioner background	(4a) Description of participating acupuncturist (qualification or professional affiliation, years in acupuncture practice, other relevant experience): Korean medical doctor administered the acupuncture treatment. He obtained license of traditional Korean Medicine and had used acupuncture in his practices for 13 years.
5. Control or comparator interventions	(5a) Precise description of the control group: There is no sham acupuncture in control group, subjects were in supine position for 15 min.

**Table 3 healthcare-08-00414-t003:** Changes on hip flexion ROM.

Unit: °	Control	Electroacupuncture	Manual Acupuncture
Baseline	67.4 (4.1)	68.0 (4.5)	62.1 (2.7)
0-min	68.6 (4.1)	71.9 (4.6) *	67.3 (3.3) *
20-min	67.8 (3.7)	72.9 (4.1) *	67.6 (3.4) *
40-min	68.1 (3.7)	72.4 (4.6) *	67.5 (3.0) *

There was treatment × time interaction (F_6,126_ = 2.91, *p* = 0.01) from 0-min post treatment to 40-min post treatment. * Regardless of treatment, there was a 5.4% increase at 0-min (*p* < 0.0001, ES = 0.43), a 5.5% increase at 20-min (*p* < 0.0001, ES = 0.47), and a 5.4% increase at 40-min (*p* < 0.0001, ES = 0.47) post treatment.

**Table 4 healthcare-08-00414-t004:** Changes in knee joint function. Values are mean (95% confidence intervals).

	Replications at 15°			Replications at 45°		
	Control	Electroacupuncture	Manual Acupuncture	Control	Electroacupuncture	Manual Acupuncture
Baseline	31.2 (4.2)	13.0 (2.7)	12.8 (3.7)	8.4 (2.1)	6.6 (1.9)	8.5 (3.0)
0-min	12.2 (3.8)	8.4 (2.0)	12.5 (3.5)	7.6 (2.2)	6.7 (1.8)	7.6 (3.2)
20-min	13.3 (4.7)	9.6 (2.3)	11.7 (3.1)	6.2 (2.3)	6.5 (1.2)	6.2 (1.3)
40-min	13.6 (4.4)	8.7 (2.0)	12.0 (3.9)	7.0 (2.6)	6.7 (1.6)	7.0 (2.1)

There was no condition × time interaction (15°: F_6,126_ = 1.53, *p* = 0.17; 45°: F_6,126_ = 1.47, *p* = 0.19).

**Table 5 healthcare-08-00414-t005:** Changes in quadriceps function. Values are mean (95% confidence intervals).

	Strength (N·m/kg)			Activation (Central Activation Ratio)		
	Control	Electroacupuncture	Manual Acupuncture	Control	Electroacupuncture	Manual Acupuncture
Baseline	3.9 (0.4)	3.7 (0.3)	3.4 (0.3)	0.95 (0.02)	0.95 (0.02)	0.97 (0.01)
0-min	3.7 (0.4)	3.4 ^†^(0.3)	3.3 (0.3)	0.95 (0.02)	0.95 (0.02)	0.96 (0.02)
20-min	3.7 (0.5)	3.5 (0.5)	3.3 (0.3)	0.94 (0.02)	0.95 (0.02)	0.96 (0.02)
40-min	3.6 (0.5)	3.5 ^‡^ (0.5)	3.1 ^§^ (0.3)	0.94 (0.02)	0.95 (0.02)	0.96 (0.01)

There was condition × time interaction for strength (electro at 0-min: 9.2%, ES = 0.60; electro at 40-min: 7.3%, ES = 0.55; manual at 40-min: 8.7%, ES = 0.54). ^†^ Different from the 0-min values in the electroacupuncture treatment (*p* < 0.0001). ^‡^ Different from the 40-min values in the electroacupuncture treatment (*p* = 0.005). ^§^ Different from the 40-min values in the manual acupuncture treatment (*p* = 0.01).

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
