# Peer review of "Electroacupuncture and Manual Acupuncture Increase Joint Flexibility but Reduce Muscle Strength"

_healthcare, 2020, doi:10.3390/healthcare8040414_

Round 1
Reviewer 1 Report
Title:
2-3 I know a lot of people refer to electroacupuncture as ‘electro’. For myself, I would prefer:
Electroacupuncture and manual acupuncture increase joint flexibility but reduce muscle strength
But this is simply my personal preference!
For ‘Electroacupuncture’, you could use either ‘Electroacupuncture’ or ‘Electro-acupuncture’, but not (in my view) ‘Electro acupuncture’.
4-5 Running Head: ‘Acupuncture effect on joint and muscle function’ would be a more complete sentence.
33 ‘to’ or ‘applied to’, rather than ‘with’ acupoints
35 why include: (flexion or extension?)
54 Instead of ‘low back [15] and hand’. How about ‘low back [15] and hand conditions’?
Table 1: Much clearer now!
Table 2: Excellent!
164 Thank you for clarifying that the EA stimulator you used was in fact symmetrical and so charge-balanced [producing a biphasic or alternating current is no guarantee that there is no residual charge imbalance!]
167 Why not state that these parameters were used in the studies you cite in your cover letter?
205 I am not a statistician, but it always makes me feel a little uneasy when people use a mix of parametric and non-parametric tests on data that is very similar except for distribution. My personal preference would be to use non-parametric on everything … but I know many researchers would not like that!
I think what you have ended up doing is fine.
Thank you for clarifying for me the meaning of ‘F6,126’ etc. I need to learn more stats!
251-268 This is clearer now. Thank you.
308 Our method to select acupoints were WAS based on individual physical condition
Reviewer 2 Report
The paper has greatly improved, and the authors' responses to my previous comments are satisfactory. I would like the clarification on two issues:
[Comment #2] Line 62-74: To adequately test the all stated hypothesis, I think it would have required two additional treatment group arms.
[Authors' response #2] If fifteen individuals in the individual acupuncture group were divided into acupuncture options after physical examination, the number of groups in 9 acupoints options would have to be met by at least ten subjects each. In that case, since the control group and the standard acupuncture group also needed ninety subjects, it is hard to subdivide groups.
I do not quite understand their explanation. If adding additional groups, the sample size would have to be recalculated (CONSORT Multi-Arm RCT Extension). Nonetheless, this study was conducted without standardized acupoints groups. The authors emphasized the potential importance of the individualized treatment approach based on the physical examination. While it might be clinically relevant, we cannot determine the superiority (or inferiority) of their individualized acupoints treatment without comparing it with a standardized acupoint group(s). (vs MA w Standardized points and EA w Standadaized points)
The related statement in the Conclusion should be removed (Line 335-336).
[Comment #3] Subject allocation and randomization procedures are not clearly stated. If the 45 subjects were allocated with a simple randomization method, it should have been n=15, each group (as per the Power analysis). However, the subjects for each group were n=13, 18, and 14. It looks like the subjects were allocated manually into separate groups by the acupuncturist based on his subjective assessments before the randomization. Have you incorporated stratification or blocking? Please clarify the random allocation process and modify the flow diagram in Figure 1.
[Authors’ response #3] The reviewer brought a valid point here. Considering dropping out of the study because of the acupuncture treatment and the superimposed technique, our stratified randomization was initially planned with a sample size of 60, not 45. Therefore, the number of subjects in each group were different. It was fortune that no one dropped. We thought that further data collection would have not changed the results when we obtained the statistical results from the current data (n=13, 18, and 14). Another reason that we stopped collecting data was based on the sample size calculations (n=13) in other previous studies (Park 2013;). We have revised the statement of our sample size calculation (Line 201-202).
Subject allocation and randomization procedures are still not very clear. The authors are now stating that the sample size was initially 60, not 45. Then, why is the flow chart starting at 50? What happened between 60 and 50?
If any stratification or blocking processes took place before the randomization based on the assessments, it should be illustrated in the flow chart accordingly.
Please also explain why did the Control group require the acupoints assignment since they were just resting in supine?
Round 2
Reviewer 1 Report
- (ROM), and knee joint
22-24. Post-treatment, both electroacupuncture and manual acupuncture decreased quadriceps strength at 0-min (electroacupuncture: 9.2%, p<0.0001, ES=0.60) and 40-min (electroacupuncture: 7.3%, p=0.005, ES=0.55; manual acupuncture: 8.7%, p=0.01, ES=0.54).
- in healthy populations. While …. as a superior treatment to manual acupuncture for pain
- randomized controlled laboratory study with repeated measurements on time.
'over time' better? I'm not sure why 'time' is needed at all: if something is repeated, time is implicitly involved!
- Ten additional participants were included when performing stratified randomization, based on considerations of drop-out rate.
252-270. In previous study has Previous studies have shown that there are at least two possible mechanisms. In the first mechanism, is known as the serotoninergic fibers, which in this 259 case from the [name?] nucleus is released of enkephalin, which hinders the transmission of nociceptive signals [31]. 260 Moreover, In the second mechanism, so-called diffuse noxious inhibitory control (DNIC), in which its implicated by 261 is the result of noxious stimuli applied to shortened muscles [31]. Regional effects may be attributed to an increase in tissue elasticity due to micro-stretching [32] and circulatory effects [33] in the constructed contracted fibers [2]. More specifically, needle insertion may stimulate release of vasoactive substances (e.g., substance P and bradykinin), 264 due to needle insertions resulting in vasodilation, which may have led to and so an increase of in blood flow [34]. Furthermore, according to Langevin [35], stated that there may be a relationship between the mechanisms of acupuncture treatment and the traditional meridians which connect the various parts of the body, reflecting . It is indicated that 267 interrelation and interactions among blood vessel network, nerve network, and immune networks are 268 considered to be the key to disclose the mechanism of acupuncture treatment and the essence of 269 meridians [36].
[As I’ve said before, KEEP IT SIMPLE!!]
274-5. In terms of the effectiveness of manual acupuncture, it resulted in decreased tibialis anterior muscle activity [37] and wrist flexor strength [7], and no change on quadriceps activation, and reduced.
Replace with: Manual acupuncture, for instance, was found to decrease tibialis anterior muscle activity [37] and wrist flexor strength [7], but with no change in quadriceps activation.
- conditions of acupuncture treatment
321-5. Lastly, our study did not include comparison with standardized acupoints (e.g., same acupoints) since a comparison between customized and standardized acupoints was not the primary purpose. Therefore, the results of this study should not be taken to imply that using individualized acupoints (based on physical examination) is superior to using standardized acupoints.
- ‘the physical examinations’: use ‘physical examination’
330-1. there does not seem to be a superior treatment between electroacupuncture and manual acupuncture
Replace with: …, neither electroacupuncture nor manual acupuncture gave a clearly superior result.
331-2. Try this: While the use of customized acupoints in patients within the same pathological conditions has not as yet been scientifically validated, our data support this common clinical practice.
Reviewer 2 Report
I am happy to see that the authors have modified the Limitation and Conclusion sections appropriately. The flow chart is still somewhat puzzling to me. However, I would leave it to the editors at this point. Overall the study is well done.
Author Response
I am so glad to hear that you are happy with our effort to complete the revised manuscript.
Thanks to your comments and suggestions that we could be done by a much higher quality of our manuscript.
Thank you,
Best regards,
Daeho Kim, Sein Jang, and Jihong Park
This manuscript is a resubmission of an earlier submission. The following is a list of the peer review reports and author responses from that submission.
Round 1
Reviewer 1 Report
Major issues
Introduction:
Line 56-58: “However, previous studies have standardized this treatment option (e.g., same acupoints), which may have confounded the results.” I am wondering if the authors realize that the usage of the individualized acupoints was one of the major confounders in this study. This issue should be included in the Limitation section.
Objectives of the study
Line 62-74: To adequately test the all stated hypothesis, I think it would have required two additional treatment group arms.
Conclusion
Line 314-315: “The acupuncturist needs to conduct a physical examination for accurate treatment to increase joint ROM.” The presented results do not support this statement.
Randomization
Subject allocation and randomization procedures are not clearly stated. If the 45 subjects were allocated with a simple randomization method, it should have been n=15, each group (as per the Power analysis). However, the subjects for each group were n=13, 18, and 14. It looks like the subjects were allocated manually into separate groups by the acupuncturist based on his subjective assessments before the randomization. Have you incorporated stratification or blocking? Please clarify the random allocation process and modify the flow diagram in Figure 1.
Why did the Control group require the acupoints assignment?
2.5. Treatments
Can you please describe further regarding EA applications? For the subjects who required all nine points (Option A, B, and C), which points were connected with the electrodes, and in which lying position(s)? Also, please add the description regarding the Control condition.
Other issues
The authors should modify the term “traditional acupuncture.” The “traditional acupuncture” generally implies the method of acupuncture based on traditional Chinese medicine principles. The term such as “manual acupuncture” would be an adequate term for the non-EA technique used in this study.
Line 121-124: Have you performed the Thomas test (original) or the Modified Thomas test? Figure 2 indicates the latter. Please specify.
Table1: So, there were six different acupoints combinations based on orthopedic assessments. Who invented this particular protocol? Are there any prior studies that showed the clinical efficacy of this particular acupuncture approach?
Also, I can understand the relationships between the modified Thomas test with ST, also the Gluteal medius with GB. However, I do not understand the rationale of using SP points for subjects with the restricted SLR test. The BL or KI points would be a more sensible choice from the meridian and neuroanatomy/segmental reflex perspectives. Please explain and state your rationale.
I advise the authors that they review the STRICTA (CONSORT Extension) checklist and include additional details regarding the acupuncture procedure.
Line 150: Please state your rationale for using 170 Hz EA frequency. The frequency seems quite high for non-analgesic EA. The authors are suggesting the increase of muscle blood flow as one of the potential mechanisms behind the increase of ROM. For the blood flow, however, it has been more commonly applied with the lower frequency. EA with 1-2 Hz has been shown to induce muscle pumping and alter the intramuscular hemodynamics. (e.g., Kubota et al. https://doi.org/10.1136%2Facupmed-2017-011433).
The terms such as “de-qi” and “meridian” abruptly appear in the main text. I generally advise the authors that they avoid using TCM jargon in scientific journals. If you are using, however, please add a short definition to explain the meaning of each term.
Line 296-298: “We had no sham-acupuncture group to test the potential placebo effects. However, it has been proved that sham-acupuncture is not different from real acupuncture [31]”. The word “proved” is a very strong word in science. If the statement is, in fact, “proven,” your whole arguments (blood flow, Substance P, CGRP, etc.) are meaningless as it implies that the study evaluated the magnitude of the placebo effect induced by different types of needling. The systematic review findings cast doubt on the validity of point locations based on the traditional acupuncture theories. Also, it might be worthwhile to state that certain types of so-called “sham acupuncture” (e.g., superficial acupuncture) are not regarded as physiologically inert. Nonetheless, I would recommend rewording the sentence as the statement is somewhat misleading.
Reviewer 2 Report
EA (electroacupuncture) & MA (manual acupuncture) increases joint 2 flexibility but reduces muscle strength
[numbers refer to lines in text]
2 traditional-acup – remove hyphen; verb plural, not singular
3 verb plural, not singular
17 using muscle tests to select acupoints is an interesting strategy
20 all these hyphens are not needed
24 interesting that quads strength decreased – but it’s in a way obvious that flexibility will increase and muscle strength decrease at the same time. What would be more interesting if one or other or both effects were persistent, and for how long.
34 Do you mean knee, not hip?! This needs correction.
36 There is no mention of acupuncture having a ‘counterirritant effect for gating pain perception’ in ref [8]. This needs careful correction.
39 BAD ERROR: shenmen does NOT refer to HE7 but to the ear point. PLEASE BE MORE CAREFUL
40 why ‘e.g.’ deqi? Are you saying deqi is the result of insertion and/or manipulation? Unclear.
41 There is NO mention of alpha motoneuron excitability in ref [5]. This needs careful correction.
46 That 'summative effect' is an unjustified assumption; results may not be simply additive.
47 1934? Please check other sources and revise – EA has been in use since the 1820s!
48 Your literature review is inadequate: there have been MANY studies comparing EA and MA, not just ‘a couple’ (language too colloquial)
49-51 These two studies are very different. One uses segmentally associated points with 40 Hz stimulation, the other used 4 Hz stimulation at non-segmentally related points. You cannot draw a general conclusion on the basis of 2 such studies!
55 ‘musculoskeletal’: this is a very limited view of oriental medicine!
58 Again, you are generalising on the basis of only 2, very different studies. You must improve the quality of the research behind your writing.
72 The reference (Gabler et al.) mentions ‘disinhibition’ of motorneurons and increasing their excitability, not their direct ‘activation’. Are these the same as ‘activation’?
81 Ethics – good
87 short-term follow-up good, but why not use the opportunity for longer-term follow-up?
Did you consider participants’ attitudes to EA and MA, whether they were ‘traditionalists’ or ‘technopohiles’?
89-90 I like the double randomisation!
Fig 1 No drop-outs! Excellent
Fig 1 No explanation of A, B, C until later in your paper. This makes life difficult for the reader. And if (as I understand) you are creating subgroups, there are too few in any group to draw meaningful conclusions
100 ‘performed physical examinations to determine the acupoints’ – HOW?! At least say ‘(see below)’ so the reader doesn’t feel completely lost.
100-104 Good blinding!
Figure 2 Did you obtain participant permission for photos?
141 Needle depth: is it possible to be more precise about range of depths used at each point?
Table 1 This is important information, but what I don’t see is any justification for the point selection on the basis of the tests conducted. You need some EVIDENCE to support this method of point selection, which will not be familiar to many acupuncture practitioners. Just providing a list of segmental innervations is not sufficient.
Many of the entries under ‘Segmental innervations’ are confusing. For instance, under (A), why do we need to know about the malleolus? And under (C), why do we need to know about web spaces between the toes?
Fig 3 You are mixing treatments, using some points without electrical stimulation in the EA groups. This is confusing. How do you know that using EA distally to a point stimulated only with MA doesn’t have a different effect to using EA alone, or using EA proximally and MA distally? Justify procedure.
It is not completely clear from the text how pairs of points were connected when EA was used. This should be made clearer.
Do you know whether using 1, 2 or 3 pairs of points affects results?
148 Was the stimulator charge-balanced, or was one needle in each pair ‘more positive’ than the other? In Fig 3, there is no consistency about whether the proximal needle received a red or a black clip.
What is the waveform produced by the device?
150 Parameters and duration are not supported by evidence! Why 170 Hz? Why only 8 minutes? Personally, I would consider that a sub-optimal treatment. From the literature, 20 minutes would be more usual.
What does ‘continuous current’ mean? Do you mean continuous mode?
151 Mention comfort level. Was this comfortable or not for participants? In particular, was MA more or less comfortable than EA?
152 What does ‘crossed the channels’ mean? I think I understand, but this is confusing. Are you talking about device channels, or acupuncture channels/meridians?
156/7 Three trials: this in itself could constitute a treatment!
170 What is the name of the device used to provide the ‘supramaximal exogenous electrical stimulus’.
How do you know that such a strong (if brief) stimulus after acupuncture didn’t in some way adversely affect participants ‘primed’ by receiving EA (much gentler?), or that those receiving EA were so traumatised by the baseline activation test that they reacted adversely to EA as a result?
There are too many unasked/unanswered questions here to be able to draw conclusions.
180 Can you justify using a TENS study as the basis for power analysis in an EA study?
183 You used mean and SD. Should you have used median and IQR instead? The sample is small. You may want to check your findings using those instead. Do your conclusions change?
186 Good to include ES! But what is SAS? I don’t know that measure …
190 I am not familiar with this way of reporting treatment x time interaction. Please explain ‘F6,126’ etc.
Table 4 I do not see any asterisks in the Table itself. Please include.
226f Again, I think the use of subgroups with different numbers of acupoints is quite confusing and does not help when you are trying to draw conclusions
237/238 Would there be a difference between two sub-optimal treatments?!
238-253 You are trying to include too much. It is always possible to create explanations from the literature, but your argument is not clear. Why bring Langevin in here, why mention the pain study again? Why not focus on the blood flow hypothesis and try and support that? Keep it simple.
268 Interesting, the time difference between MA and EA
278 I admire your belief that acupoint selection should be based on the results of physical examination, but you need to justify the link between them on the basis of literature review or other evidence. In fact, why not do a study just on this?
289-291 Good that you acknowledge the treatment was brief, and also the need for more than 1 session
297-8 ‘it has been proved that sham-acupuncture is not different from real acupuncture’. I don’t think that’s exactly what Moffet said!
